# Sex differences in the applicability of Western cardiovascular disease risk prediction equations in the Asian population

Hee-Sook Lim[1], Hyein Han[2], Sungho Won[2,3,4,5], Sungin Ji[6], Yoonhyung Park[6‡]*, Hae-Young Lee[7,8‡]*

1 Department of Gerontology, AgeTech-Service Convergence Major, Graduate School of East-West Medical Science, Kyung Hee University, Yongin, Republic of Korea, 2 Department of Public Health Sciences, Seoul National University, Seoul, Republic of Korea, 3 Interdisciplinary Program in Bioinformatics, Seoul National University, Seoul, Republic of Korea, 4 Institute of Health and Environment, Seoul National University, Seoul, Republic of Korea, 5 RexSoft Corps, Seoul, Republic of Korea, 6 Department of Preventive Medicine, Soonchunhyang University College of Medicine, Cheonan, Republic of Korea, 7 Department of Internal Medicine, Seoul National University College of Medicine, Seoul, Republic of Korea, 8 Department of Internal Medicine, Seoul National University Hospital, Seoul, Republic of Korea

☯ These authors contributed equally to this work.
‡ HYL and YP also contributed equally to this work as correspondence author.
* hylee612@snu.ac.kr (HYL); parky@sch.ac.kr (YP)

**Data Availability Statement:** Third party data was obtained for this study from National Health Insurance Service-National Sample Cohort (NHIS-NSC), South Korea. Data may be requested from

## Abstract

### Aims

Cardiovascular diseases (CVDs) are the most common cause of death, but they can be effectively managed through appropriate prevention and treatment. An important aspect in preventing CVDs is assessing each individual's comprehensive risk profile, for which various risk engines have been developed. The important keys to CVD risk engines are high reliability and accuracy, which show differences in predictability depending on disease status or race. Framingham risk score (FRS) and the atherosclerotic cardiovascular disease risk equations (ASCVD) were applied to the Korean population to assess their suitability.

### Methods

A retrospective cohort study was conducted using National Health Insurance Corporation sample cohort from 2003 to 2015. The enrolled participants over 30 years of age and without CVD followed-up for 10 years. We compared the prediction performance of FRS and ASCVD and calculated the relative importance of each covariate.

### Results

The AUCs of FRS (men: 0.750; women: 0.748) were higher than those of ASCVD (men: 0.718; women: 0.727) for both sexes (Delong test P <0.01). Goodness of fits (GOF) were poor for all models (Chi-square P < 0.001), especially, underestimation of the risk was pronounced in women. When the men's coefficients were applied to women's data, AUC (0.748; Delong test P<0.01) and the GOF (chi-square P = 0.746) were notably improved in FRS. Hypertension was found to be the most influential variable for CVD, and this is one of

NHIS after creating an account and registering for access. More access information can be found on the NHIS website (https://nhiss.nhis.or.kr). The authors confirm that interested researchers would be able to access these data in the same manner as the authors. The authors also confirm that they had no special access privileges that others would not have.

**Funding:** This study was supported by the Korean Society of Cardiovascular Disease Prevention (Grant number, 2021-01, HY Lee).

**Competing interests:** The authors have indicated that they have no conflicts of interest in the content of this article.

the reasons why FRS, having the highest relative weight to blood pressure, showed better performance.

## Conclusion

When applying existing tools to Korean women, there was a noticeable underestimation. To accurately predict the risk of CVD, it was more appropriate to use FRS with men's coefficient in women. Moreover, hypertension was found to be a main risk factor for CVD.

## Introduction

Cardiovascular disease (CVD) is the most common cause of death except overall cancers, accounting for 26.9% of all deaths in Korea [1]. Currently, 11.27 million Korean patients are taking medications for high blood pressure, diabetes, or dyslipidemia [2]. In addition, the burden of chronic diseases is increasing due to the growing population of older people and the unhealthy lifestyle factors that exacerbate CVD [3]. In an attempt to reduce the incidence of CVD, early intensive prevention strategy based on individual risk prediction is necessary.

Various CVD risk engines have been developed to predict CVD. The Framingham risk score (FRS) has been used to evaluate the risk of coronary heart disease (CHD) [4]. In addition, the American Heart Association (AHA) developed the atherosclerotic CVD (ASCVD) risk score, which broadened the relevance of risk engines within different ethnic groups [5, 6]. However, the applicability of the Western risk engines to non-White, non-African American races has been debated [7, 8]. A study on adults who visited 18 health examination centers revealed that the FRS overestimated the risk of CHD in Korean population with a low incidence of CHD [9]. Among men who participated in the Korean Heart Study, the risk calculation by the ASCVD risk score was reported to have overestimated CVD risk [10]. The risk index for predicting CVD varies according to race, gender, and other factors. In addition, with changes in the lifestyle and advances in the development of chronic-disease treatment methods; previous prediction indexes may no longer be applicable.

Korea is experiencing longer life expectancy, particularly among women, and chronic diseases such as stroke and ischemic heart disease are a major problem. To decrease the incidence of complications associated with chronic diseases, reliable predictive tools are needed, but there has been limited research on predictive risk indexes, and no model suitable for Korean has been developed. The National Health Insurance Service-National Sample Cohort (NHIS-NSC) database is now available to generate long-term and more reliable risk assessments than in the past. Therefore, using the NHIS-NSC data, this study compared the prediction performance of the FRS and the ASCVD risk score and identified risk factors for CVD in South Korea.

## Materials and methods

### Study population

In this study, we used the NHIS-NSC database. Korea has a single, government-maintained NHIS, and the universal NHIS provides free biennial health examinations to eligible NHIS members aged ≥ 40 years. The cohort data include medical services claim data, and pharmacy claim data [11]. Korean NHIS has potential for big data analysis because it is a unified insurance system covering > 90% of Koreans; therefore, nearly the entire population's use of

medical resources can be examined by the claim data. Moreover, the Korean doctors have relatively discretionary authority in medical decision-making and treatment; therefore, NHIS data facilitates the comparison of the effect of various diagnostic modalities and treatment strategies.

From the NHIS-Health Screening Cohort between 2002 and 2015 [12], individuals with a history of hypertension or type 2 diabetes mellitus (T2DM) in 2002 and those with any missing health-screening data were excluded, resulting in a total of 117,926 participants. The inclusion criteria for the current analyses were chosen to match those used in the development of the pooled cohort risk equations. Then, participants aged under 30 years ($n$ = 18,702), those diagnosed with CVD ($n$ = 13,054), and those exhibiting outliers ($n$ = 1,430) were identified. Outliers were defined as follows: total cholesterol > 300 mg/dL, high-density lipoprotein (HDL)-cholesterol > 100 mg/dL, body mass index (BMI) > 100 kg/m$^2$, low-density lipoprotein (LDL)-cholesterol > 1,000 mg/dL, or triglycerides > 1,500 mg/dL. Finally, we included patients who were still alive 10 years after the baseline date. Consequently, data from 84,087 participants were available, including 50,619 men and 33,468 women. The FRS and the ASCVD evaluations also specify inclusion criteria, and 77,396 (50,606 men vs. 26,790 women) and 58,304 (33,158 men vs. 25,146 women) participants fulfilled the criteria for each, respectively (Fig 1). This study was approved by the National Health Insurance Service (Approval No. NHIS-2019-2-265) and the Institutional Review Board (IRB) of Soonchunhyang University (IRB No. 201907-BM-044-01) in accordance with the Declaration of Helsinki.

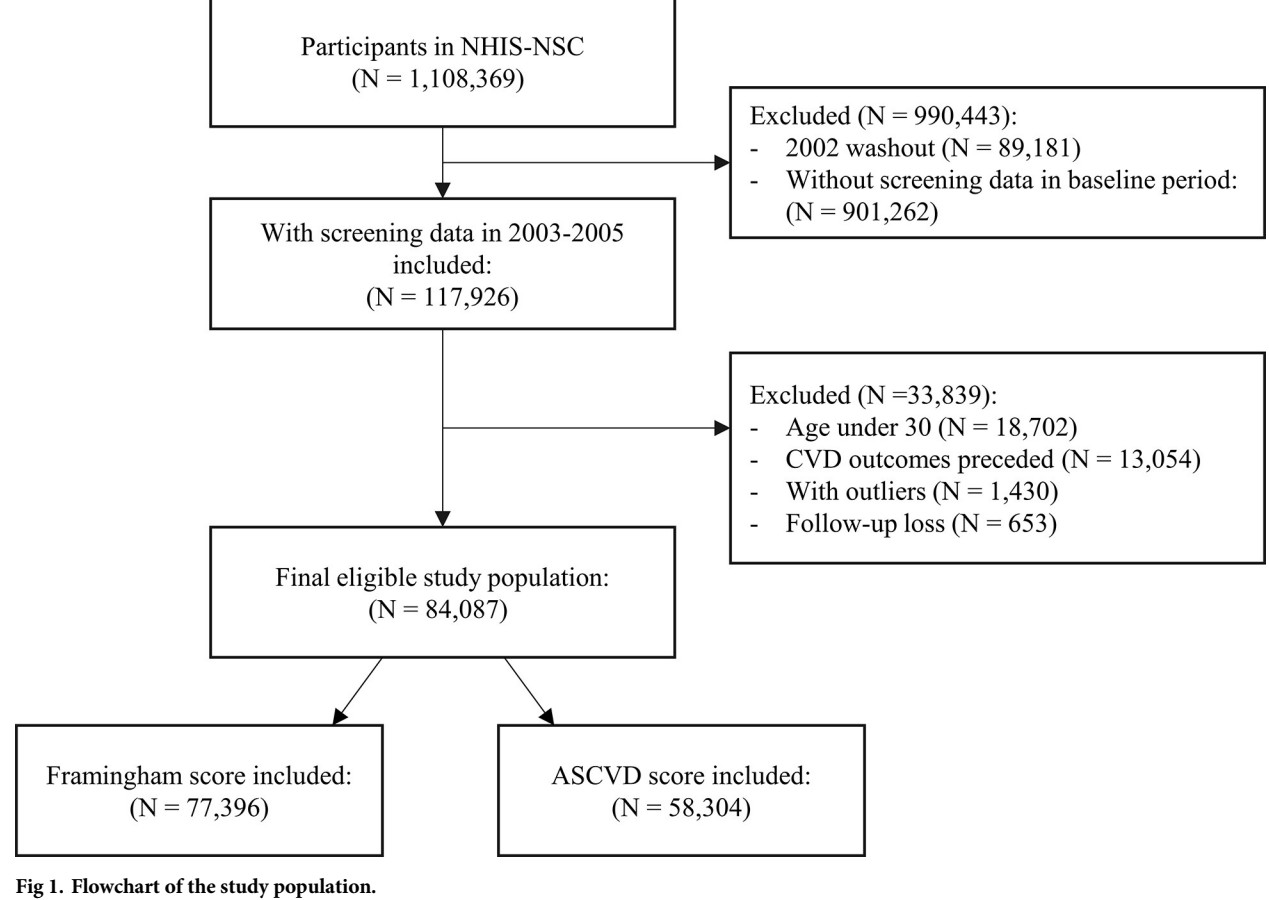

**Fig 1. Flowchart of the study population.**

## Data collection

Diagnoses of subjects in this study were confirmed by linking the NHIS-NSC data with the chronic disease descriptions and the International Classification of Diseases (ICD) 10th codes. Medical examination information was collected from the NHIS (between 2003 and 2005), and from records of health examinations during the transition period as well as from cancer screening data. Sex, age, total cholesterol, HDL-cholesterol, blood pressure, recent treatment for hypertension, T2DM, and smoking data were used to calculate the FRS and the ASCVD risk score. In addition, BMI, LDL-cholesterol, and triglyceride levels were examined as potential risk factors. T2DM was defined using the ICD 10th code. E11.9 or fasting serum glucose level > 126 mg/dL. Untreated hypertension was defined as systolic blood pressure (SBP) > 140 mmHg or a history of hypertension diagnosis (ICD code = I10). The CVD outcomes in this study were defined 10 years after the baseline data collection as the occurrence of ischemic heart disease (I20–21), coronary heart disease (I48, 50), cardiac arrest (I46), hemorrhagic stroke (I60–I62), or ischemic stroke (I63–I64, G45).

## Risk score calculation

Both the FRS and the ASCVD were developed based on the Cox proportional hazards method, and the features used in them are nearly identical [6, 13]. Equation parameters are listed in S1 Table. For example, CVD risk for men in FRS is calculated as follows:

$$L = \beta_0 \times ln(Age) + \beta_1 \times ln(Total\ cholesterol) + \beta_2 \times ln(HDL\ cholesterol) + \beta_3 \times ln(Systolic\ blood\ pressure) + \beta_4 \times Treated\ for\ blood\ pressure + \beta_5 \times Smoker + \beta_6 \times Diabetes - Mean\ (Coefficient \times Value)$$

$$CVD\ Risk = 1 - (Baseline\ survival)^{exp(L)}$$

Values in S1 Table corresponding to each variable should be inserted in each beta.

## Statistical analyses

Baseline characteristics were compared between sexes using a t-test for continuous variables and a chi-square test for categorical variables. Thereafter, we evaluated and calibrated the FRS and the ASCVD risk score. The accuracy of the predicted outcomes was assessed by calculating the area under the curve (AUC), and the AUC values between models were compared using the Delong test [14]. Goodness of fit (GOF) for each model was evaluated by Hosmer-Lemeshow test by comparing predicted risks and the actual risks [14]. The chi-square values were estimated, and a calibration plot was created to identify risk overestimates. We included LDL-cholesterol, triglycerides, fasting serum glucose, and BMI as additional predictors, and used the Cox proportional hazards method to build a data-driven prediction model, which was considered as one of the reference tools when evaluating given models. We employed 5-fold cross-validation and explored all possible combinations of covariates to identify the best combination with the highest AUC. We evaluated the relative importance of each covariate by calculating the relative proportions of variances with all but one covariate. The statistical significance was set at 0.05 and R software (version 3.3.3; The R Foundation for Statistical Computing, Vienna, Austria) was used.

## Results

### Descriptive statistics

Several distinctive features were observed between men and women (Table 1). Women (46.3 ± 9.9) were older than men (42.5 ± 10.1), and men had a higher smoking rate (44.8% vs.

**Table 1. Baseline characteristics of the study population.**

| Risk factors | Men | Women | P |
|---|---|---|---|
| | (N = 50,619) | (N = 33,468) | |
| Age, year (Mean ± SD) | 42.5 ± 10.1 | 46.3 ± 9.9 | <0.001 |
| Body Mass Index, kg/m$^2$ (Mean ± SD) | 23.7 ± 2.8 | 22.9 ± 2.9 | <0.001 |
| Fasting serum glucose, mg/dL (Mean ± SD) | 92.4 ± 19.4 | 89.9 ± 17.2 | <0.001 |
| Total cholesterol, mg/dL (Mean ± SD) | 193.1 ± 32.3 | 191.5 ± 34.0 | <0.001 |
| LDL-cholesterol, mg/dL (Mean ± SD) | 115.2 ± 35.8 | 119.2 ± 34.7 | <0.001 |
| HDL-cholesterol, mg/dL (Mean ± SD) | 51.9 ± 12.3 | 58.0 ± 13.0 | <0.001 |
| Triglyceride, mg/dL (Mean ± SD) | 149.0 ± 97.0 | 107.7 ± 64.4 | <0.001 |
| Systolic BP, mmHg (Mean ± SD) | 123.4 ± 14.9 | 117.8 ± 16.6 | <0.001 |
| Smoking, n (%) | 22659 (44.8%) | 637 (1.9%) | <0.001 |
| Type 2 diabetes[a], n (%) | 1911 (3.8%) | 1019 (3.0%) | <0.001 |
| Hypertension[b], n (%) | 3125 (6.2%) | 2374 (7.1%) | <0.001 |

SD, standard deviation; LDL, low-density lipoprotein; HDL, high-density lipoprotein; BP, blood pressure.

[a] Type 2 diabetes was defined as fasting serum glucose ≥126 mg/dL or history of diagnosis (ICD code-E11.9).

[b] Hypertension was defined as systolic blood pressure of ≥140 mmHg or history of diagnosis (ICD code-I10).

1.9%) and higher levels of triglycerides (149.0 ± 97.0 vs. 107.7 ± 64.4). The incidence rates of each subtype of CVD outcomes also varied based on sex (Table 2). Especially, the incidence of I63 (Cerebral infarction), which was the most frequently occurring subtype, was significantly higher in women (327.63 [95% CI, 308.46 to 347.58]) compared to men (240.65 [95% CI, 227.3 to 254.5]).

## Comparison of performance between FRS and ASCVD risk score

The results of predictability assessment are presented in Table 3. The AUCs of FRS (men: 0.750 [95% CI, 0.741 to 0.760]; women: 0.748 [95% CI, 0.738 to 0.759]) were significantly higher than those of ASCVD (men: 0.718 [95% CI, 0.707 to 0.729]; women: 0.727 [95% CI, 0.715 to 0.738]) for both sexes (P < 0.01).

Fig 2 displays I results of comparing the predicted and actual incidence of CVD using both the FRS and the ASCVD risk score. The overall distributions for both scores were divided into 10 deciles to present the mean predicted score for each interval. Chi-square tests produced P-

**Table 2. Incidence rates of CVD outcomes with person-years and events at 10 years of follow-up in the populations.**

| ICD code | Description | Men | | Women | | P |
|---|---|---|---|---|---|---|
| | | Events | Incidence rate[a] (95% CI) | Events | Incidence rate[a] (95% CI) | |
| I63 | Cerebral infarction | 1203 | 240.65 (227.3, 254.5) | 1078 | 327.63 (308.46, 347.58) | <0.001 |
| I61 | Intracerebral hemorrhage | 125 | 24.73 (20.64, 29.32) | 80 | 23.94 (19.07, 29.57) | 0.876 |
| G45 | Transient cerebral ischemic attacks and related syndromes | 142 | 28.09 (23.72, 32.97) | 174 | 52.11 (44.75, 60.24) | <0.001 |
| I50 | Heart failure | 74 | 14.63 (11.54, 18.22) | 80 | 23.93 (19.06, 29.56) | 0.003 |
| I48 | Atrial fibrillation and flutter | 541 | 107.44 (98.64, 116.75) | 301 | 90.38 (80.55, 100.98) | 0.018 |
| I20 | Angina pectoris | 591 | 117.42 (108.21, 127.14) | 395 | 118.73 (107.4, 130.83) | 0.89 |
| I21 | Acute myocardial infarction | 450 | 89.38 (81.38, 97.9) | 271 | 81.36 (72.05, 91.43) | 0.236 |
| I60 | Subarachnoid hemorrhage | 62 | 12.26 (9.45, 15.57) | 67 | 20.04 (15.61, 25.23) | 0.006 |
| I46 | Cardiac arrest | 56 | 11.07 (8.42, 14.22) | 16 | 4.78 (2.8, 7.52) | 0.003 |
| I62 | Other nontraumatic intracranial hemorrhage | 50 | 9.88 (7.39, 12.88) | 16 | 4.78 (2.8, 7.52) | 0.013 |

CVD, cardiovascular disease; ICD, International Classification of Diseases; CI, confidence interval

[a] Incidence rate per 100,000 person-years

**Table 3. Comparison of prognostic performance between FRS and ASCVD risk score models.**

| | | Framingham | ASCVD |
|---|---|---|---|
| Men | AUC (95% CI) | 0.750 (0.741–0.760) | 0.718 (0.707–0.729) |
| | Sensitivity | 63.8 | 61.2 |
| | Specificity | 75.7 | 71.6 |
| | P-value | <0.01 [a] | |
| Women | AUC (95% CI) | 0.748 (0.738–0.759) | 0.727 (0.715–0.738) |
| | Sensitivity | 73.0 | 61.7 |
| | Specificity | 64.6 | 72.8 |
| | P-value | <0.01 [b] | |

FRS, Framingham risk score; ASCVD, atherosclerotic cardiovascular disease risk equations; AUC, area under the curve; CI, confidence interval

[a] P-values were generated from the Delong test comparing AUCs for Framingham and ASCVD in men

[b] P-values were generated from the Delong test comparing AUCs for Framingham and ASCVD in women

values less than 0.001, indicating poor GOF for all models. In men, the original FRS model predicted a CVD incidence of 7.93%, while the observed incidence was 6.01%. The ASCVD risk score, on the other hand, estimated the CVD incidence at 6.38%, but the observed incidence was 9.47%. In women, the predicted CVD incidence by the original FRS model was 3.90%, whereas the observed incidence was 6.92%. Furthermore, the ASCVD risk score estimated the CVD incidence as 2.23%, while the observed incidence was 8.76%.

These results indicate that the observed incidence of CVD in women was underestimated by both tool when using the original model. Consequently, we decided to apply the men's coefficients to the women's data.

### Application of men coefficients in women data

The AUC demonstrated a significant increase from 0.748 (95% CI, 0.738 to 0.759) to 0.755 (95% CI, 0.750 to 0.766) for the FRS (P < 0.01). This improvement was corroborated by a substantial improvement in the GOF test, with the Hosemer-Lemeshow test no longer exhibiting significance (P = 0.746). Conversely, in the case of ASCVD, there was no significant change in the AUC (P = 0.39), and the P-value from the chi-square test was lower than 0.001, indicating persistent underestimation of CVD risk (Fig 3). In summary, the FRS showed a marked improvement in model performance, when compared to the ASCVD.

### Developing the data-driven model

We developed a data-driven model that exhibited the best performance in our data, with detailed coefficients outlined in S1 Table. The model's AUC was 0.780 (95% CI, 0.771 to 0.789) for men and 0.776 (95% CI, 0.766 to 0.786) for women. Hosmer-Lemeshow test yielded P-value of 0.003 for men, 0.37 for women, respectively (Fig 4). By developing the data-customized model, we used it as a benchmark for evaluating and comparing pre-existing models. The coefficient that swapped in the FRS coefficients for women–yielding the best prediction among given models–showed performance closest to that of the data-driven model. Although it had a slightly lower AUC, it produced identical results in the GOF test.

### The impact of variables on CVD risk prediction

To discern which covariate exerted the most significant effect on the outcome, we computed the relative proportion of variances explained by each covariate (Table 4). When employing

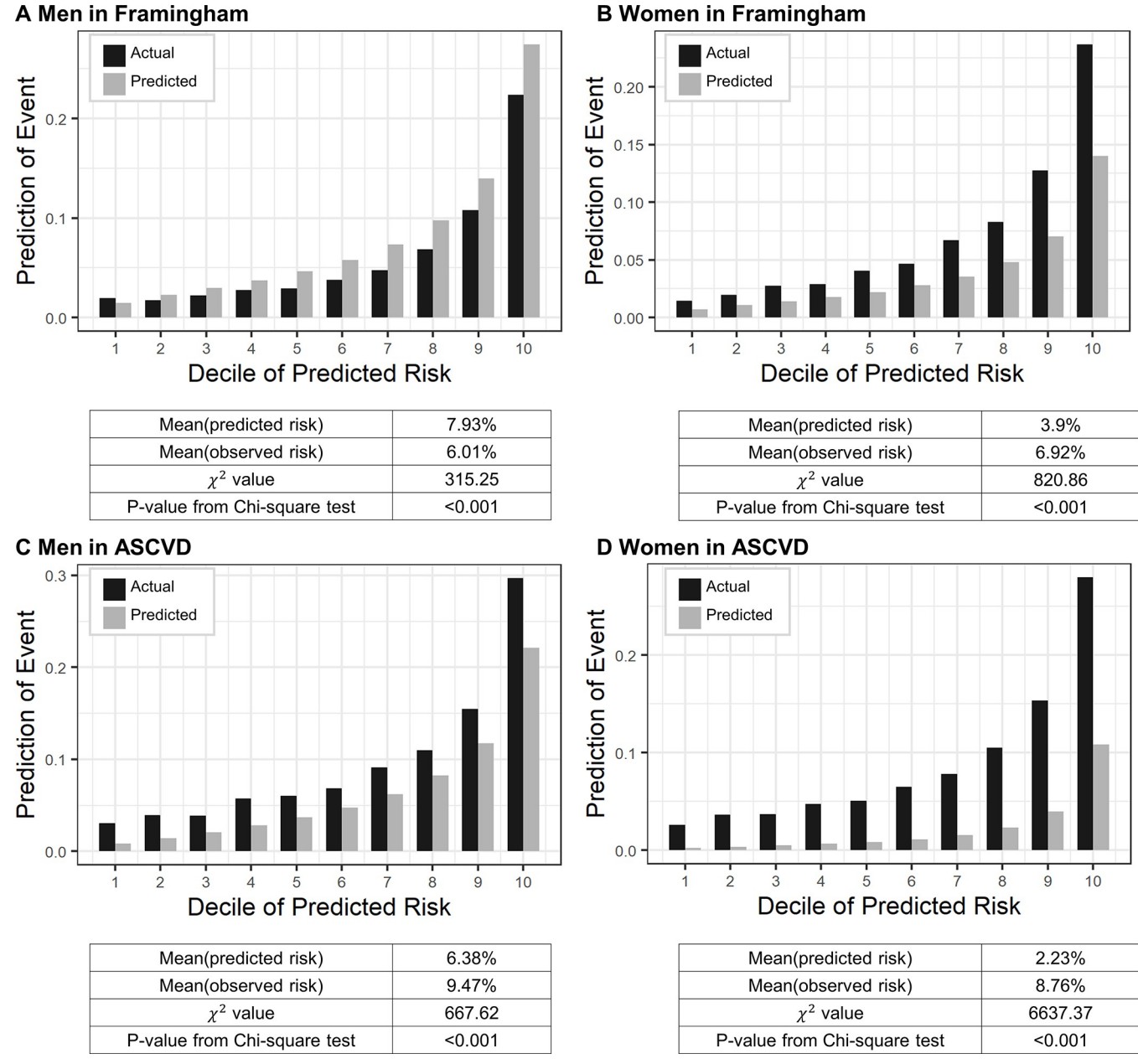

**Fig 2. Comparison of the calibration by decile between FRS and ASCVD models.** Vertical bars represent observed (black) and predicted (grey) risks.

the FRS, the most critical variable was blood pressure (Ln-treated SBP was 5.785 for men and 15.881 for women); for the ASCVD, the most important variable was smoking for men (14.658) and age for women (21.698). In the data-driven model, treated SBP emerged as the most influential variable for both men (10.891) and women (6.192). The value for 'Ln-treated SBP' was the highest in all models except for the ASCVD, indicating that the most influential variable for CVD outcome was consistent in both the FRS and the data-driven models. This alignment substantiates our finding that the FRS demonstrated higher performance and a better fit with our data, which indicates blood pressure was the most significant risk factor in our study population, and the model that attributed the greatest weight to SBP (FRS) best fit our data.

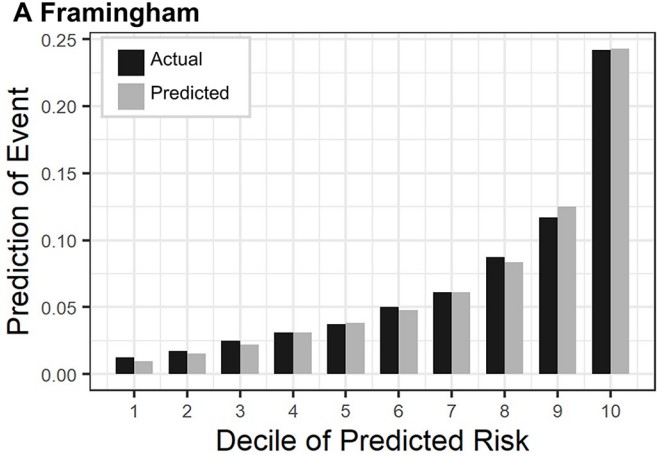

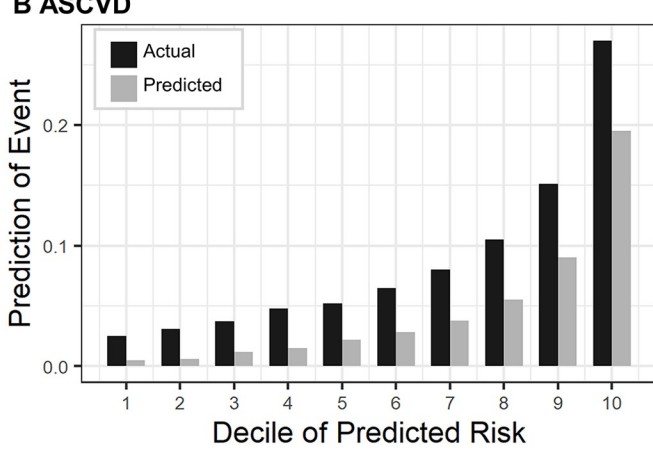

**A Framingham**

| Prediction accuracy | |
|---|---|
| AUC (95% CI) | 0.755 (0.750-0.766) |
| Sensitivity | 68.2 |
| Specificity | 70.8 |
| P-value from Delong test | <0.01 |
| **Goodness of fit** | |
| Mean(predicted risk) | 6.85% |
| Mean(observed risk) | 6.92% |
| $\chi^2$ value | 5.11 |
| P-value from Chi-square test | 0.746 |

**B ASCVD**

| Prediction accuracy | |
|---|---|
| AUC (95% CI) | 0.726 (0.715-0.738) |
| Sensitivity | 63.7 |
| Specificity | 71.2 |
| P-value from Delong test | 0.39 |
| **Goodness of fit** | |
| Mean(predicted risk) | 4.63% |
| Mean(observed risk) | 8.76% |
| $\chi^2$ value | 1600.65 |
| P-value from Chi-square test | <0.001 |

**Fig 3. Performance of model of men coefficients applied to female data.** Vertical bars represent observed (black) and predicted (grey) risks. P-values from Delong test were used to compare AUCs generated using the original model coefficients.

## Discussion

The main findings of this study indicate that FRS and ASCVD risk scores significantly underestimate the CVD risk in Korean women. Therefore, especially in the case of FRS, it was more beneficial to apply the same coefficients as those for men to improve risk prediction. We developed a data-driven model with higher weights assigned to blood pressure, which showed the best performance in our dataset. We then compared this with the given models. The better performance of FRS in our data might be attributed to the fact that, similar to the data-driven model, it placed the highest weight on blood pressure. In addition, body mass index (BMI) for men, and BMI and LDL cholesterol for women also had important role in risk calculation in the data-driven model.

After several studies showed that the FRS overestimated cardiovascular risk in large-scale cohorts of American adults, the AHA and the American College of Cardiology introduced the ASCVD risk calculator, which provided more consistent estimates and forecasts for health insurance claims [8, 15, 16]. However, overestimates were still observed in multi-ethnic studies, possibly due to the lack of active monitoring or because of other variables that are affected by race or the living environment [17, 18]. Our results are somewhat different. The incidence of CVD after 10 years among the Korean adults who initially did not have CVD was underestimated by the ASCVD risk score and overestimated by the FRS in men. Both tools produced

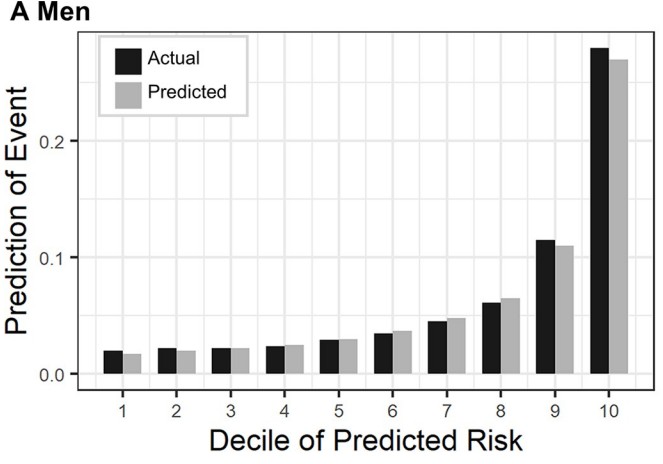

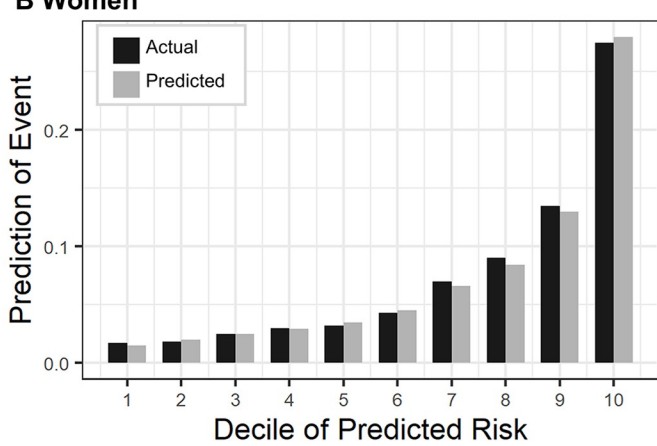

**A Men**

| Prediction accuracy | |
|---|---|
| AUC (95% CI) | 0.780 (0.771-0.789) |
| Sensitivity | 65.6 |
| Specificity | 79.3 |
| P-value from Delong test | <0.01 |
| **Goodness of fit** | |
| Mean(predicted risk) | 6.15% |
| Mean(observed risk) | 6.17% |
| $\chi^2$ value | 23.051 |
| P-value from Chi-square test | 0.003 |

**B Women**

| Prediction accuracy | |
|---|---|
| AUC (95% CI) | 0.776 (0.766-0.786) |
| Sensitivity | 67.4 |
| Specificity | 75.1 |
| P-value from Delong test | <0.01 |
| **Goodness of fit** | |
| Mean(predicted risk) | 7.08% |
| Mean(observed risk) | 7.09% |
| $\chi^2$ value | 8.675 |
| P-value from Chi-square test | 0.370 |

**Fig 4. Performance of data-driven model.** Vertical bars represent observed (black) and predicted (grey) risks. P-values from Delong test were used to compare the AUCs for generated using the original coefficients of the Framingham model.

underestimates for women, and prediction rates and observation rates differed significantly. For the risk equations to be useful in clinical trials, risk estimates must be calibrated to resemble the observed incidence of a disease. The higher observed incidence of CVD among women compared with men and the differences between predicted and observed incidences demonstrated that the existing tools need adjustment. Similar underestimates were recorded in other Asian studies [19–21].

Korea has a growing older population, with 13.8% of the population aged 65 years or older in 2017. This social ageing phenomenon is expected to increase to 43.9% by 2060 [22]. The World Health Organization also reported that South Korean women born in 2030 have a life expectancy of 90.82 years, which is much higher than in many other countries [23, 24]. The lifestyles of South Korean men and women have recently become more similar. In fact, when the existing CVD prediction tools were applied to women, while taking the greater mean weight of men into account, the prediction accuracies improved. Many studies have noted the high prevalence of chronic diseases after the menopause, including hypertension and dyslipidemia [25, 26]. We found that to predict the incidence of CVD in women more accurately, similar risk estimates to those used for men should be applied. In addition, efforts should be made to lower the incidence of CVD among women. Hypertension is a major risk factor for CVD. Therefore, identifying adults who are at high risk of having hypertension is important

**Table 4. Variance explained by each covariate for Framingham, ASCVD risk score, and the data-driven models.**

|  | Data-driven model | | Framingham | | ASCVD | |
|---|---|---|---|---|---|---|
|  | **Men** | **Women** | **Men** | **Women** | **Men** | **Women** |
| Ln age | - | - | 0.461 | 0.238 | 3.735 | 21.698 |
| Ln age square | 0.552 | 0.561 | - | - | - | 36.375 |
| Ln total cholesterol | - | - | 0.036 | 0.046 | 3.615 | 5.108 |
| Ln HDL-cholesterol | - | 0.002 | 0.048 | 0.026 | 3.619 | 9.655 |
| Ln treated SBP | 10.891 | 6.192 | 5.785 | 15.881 | 7.219 | 9.94 |
| Ln untreated SBP | 9.284 | 5.034 | 5.434 | 15.08 | 7.003 | 9.309 |
| Smoking | 0.003 | - | 0.106 | 0.005 | 14.658 | 1.107 |
| Type 2 diabetes | - | - | 0.012 | 0.014 | 0.022 | 0.015 |
| Ln body mass index | 0.004 | 0.004 | - | - | - | - |
| Ln age × Ln total cholesterol | - | - | - | - | 7.557 | 13.356 |
| Ln age × Ln HDL-cholesterol | - | - | - | - | 3.89 | 10.251 |
| Ln age × smoking | - | - | - | - | 11.564 | 0.823 |
| Ln LDL-cholesterol | - | 0.009 | - | - | - | - |

ASCVD, atherosclerotic cardiovascular disease risk equations; SBP, systolic blood pressure; HDL, high-density lipoprotein; LDL, low-density lipoprotein.

[a] Relative proportion of variances explained by each covariate over the variance of the prediction model.

"-"means N/A.

for the cost-effective implementation of interventions [27, 28]. In our the data-driven model, the most important variable for predicting CVD was blood pressure, especially treated blood pressure. Some variables, such as total cholesterol or HDL-cholesterol were excluded, but LDL-cholesterol should be included when the model is applied to women. Recently, obesity has become a serious problem in South Korea, and the prevalence of dyslipidemia among women has also increased. An analysis by the South Korean Health Insurance Review and Assessment Service showed that, compared to individuals with normal weight, men and women who were overweight or obese had 2.86-fold and 1.30-fold higher mortality rates, respectively. HDL-cholesterol measurements are frequently used with the FRS and the ASCVD models as a consistent biomarker for cardiovascular health. However, some Mendelian studies have suggested that HDL-cholesterol is not a causal cardiovascular risk factor and that high HDL-cholesterol has not been conclusively determined to lower CVD risk [29]. Blood pressure is independently associated with the risk of CVD in many studies [30]. Jee et al. [9] showed that the risk of coronary artery disease was associated with LDL-cholesterol levels in men and with high blood pressure in Korea women [31]. In 2018, the prevalence of hypertension in South Korea was 32.3% for men and 21.3% for women, whereas a lower proportion of men (48.4%) than women (65.5%) were undergoing treatment [32, 33]. Therefore, the prevalence of high blood pressure far exceeded efforts to control the problem. Therefore, more effort needs to increase treatment rate. In this regard, CVD-risk based individual care might enhance the treatment results [34].

This study's limitation may include measurement errors. Random errors may have decreased the study's power to detect associations, and systematic errors may have altered the distribution of events and, perhaps, the risk factor–disease relationships, if there are errors that are related to the exposure status. One of the major strengths of this study is that large-scale South Korean population cohort data were used without arbitrary selection of the subjects. This is a comprehensive study, it includes more variables than with many previous studies,

and its results are consistent with previous research on the relationship of CVD with both blood pressure and BMI in the South Korean population. In addition, we identified factors that required recalibration and made adjustments for variables to predict the incidence of CVD more accurately in the South Korean population. Because predicting the future is difficult and requires quantifying many factors, the optimal risk prediction model should ultimately be limited to CVD risk factors, which needs to be validated in the prospective cohort studies.

## Conclusions

Our study on the incidence of CVD in a South Korean cohort over a 10-year period showed that using the FRS and the ASCVD risk score underestimates the CVD incidence in Korean population, especially in women. As a practical solution, it would be better to apply the men's coefficients in risk engines, regardless of sex. Moreover, hypertension was found to be a main risk factor for the CVD outcome.

## Supporting information

**S1 Table. Coefficients for all models.**
(DOCX)

## Author Contributions

**Conceptualization:** Hee-Sook Lim, Hyein Han, Sungho Won, Sungin Ji, Yoonhyung Park, Hae-Young Lee.

**Data curation:** Hee-Sook Lim, Hyein Han.

**Formal analysis:** Hee-Sook Lim, Hyein Han, Sungho Won.

**Funding acquisition:** Yoonhyung Park, Hae-Young Lee.

**Investigation:** Hee-Sook Lim.

**Methodology:** Hee-Sook Lim, Sungho Won, Sungin Ji, Yoonhyung Park, Hae-Young Lee.

**Project administration:** Sungho Won, Sungin Ji, Yoonhyung Park, Hae-Young Lee.

**Resources:** Hyein Han, Sungin Ji, Yoonhyung Park.

**Supervision:** Hae-Young Lee.

**Validation:** Hee-Sook Lim, Hyein Han.

**Visualization:** Hee-Sook Lim, Hyein Han.

**Writing – original draft:** Hee-Sook Lim, Hyein Han, Sungin Ji.

**Writing – review & editing:** Sungho Won, Yoonhyung Park, Hae-Young Lee.

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
