## [Decision Letter · Decision Letter 0]

26 May 2023

PONE-D-23-01704Sex Differences in the Applicability of Western Cardiovascular Disease Risk Prediction Equations in the Asian PopulationPLOS ONE

Dear Dr. Lee,

Thank you for submitting your manuscript to PLOS ONE. After careful consideration, we feel that it has merit but does not fully meet PLOS ONE’s publication criteria as it currently stands. Therefore, we invite you to submit a revised version of the manuscript that addresses the points raised during the review process.

Even if the manuscript is of interest, the text and presentation of the data could be improved. 

We look forward to receiving your revised manuscript.

Kind regards,

Chiara Pavanello

Academic Editor

PLOS ONE

“This work was supported by the research grant from Seoul National University Hospital (#3020200110, H.-Y.L)”

4. Thank you for stating the following in the Acknowledgments/ Funding Section of your manuscript:

“Research Grant from Seoul National University Hospital (#3020200110, H.-Y.L)”

“This work was supported by the research grant from Seoul National University Hospital (#3020200110, H.-Y.L)”

“The authors have indicated that they have no conflicts of interest in the content of this article.”

Reviewers' comments:

Reviewer's Responses to Questions

**Comments to the Author**

1. Is the manuscript technically sound, and do the data support the conclusions?

Reviewer #1: Yes

Reviewer #2: No

2. Has the statistical analysis been performed appropriately and rigorously? 

Reviewer #1: I Don't Know

Reviewer #2: No

3. Have the authors made all data underlying the findings in their manuscript fully available?

Reviewer #1: Yes

Reviewer #2: Yes

4. Is the manuscript presented in an intelligible fashion and written in standard English?

Reviewer #1: Yes

Reviewer #2: No

5. Review Comments to the Author

Reviewer #1: Hee-Sook Lim et al. performed a good work evaluating the prediction accuracy of the FRS and the ASCVD in the Korean population.

I found this study interesting as it provides a new cardiovascular risk score model which seems more suitable for Koreans, allowing sex differences in cardiovascular risk estimation and providing a practical solution.

English language is fluent, concepts are expressed in a clear and easily understandable way.

Reviewer #2: The study aims at providing comparison between equations for the assessment of ACVD risk in a very large registry representative of the Korean population. The purpose of the study, at least as it can be derived from the main information given by authors, is valuable but there are important flaws that affect the validity of this submission:

1) Abstract. The introduction is poorly written and needs a very extensive rephrasing in order to provide to the reader a very straightforward message. Actually, there is not a clear rational about the gender-difference that is postulated in the heading and the design of the study and the analysis.

2) Abstract: conclusion section is even more elusive and very poorly clear. At the end the reviewer believes that the reader would not be provided of a clear message from this data. Please, provide a very important revision of the abstract and, consequently, of the rationale in the background/introduction section.

3) Appendix 1. The descriptive table is not very helpful and actually the reviewer is not convinced about its added value. Vice versa, a table with descriptive statistics of the population strata according to CVD risk would be even more informative. In this way, this could be the first main table starting from the results section.

4) Table1 and Table 2. The way covariates are presented is not easy-to-get for the readers. Perhaps, some more explanation of how the models are conducted a which is the sense of each factor included in this table would be appreciated.

5) Figure 2, Figure 3, Table 3. Major point of criticism: actually, the authors are evaluating the areas under the curve of each model and how the performance is, for each single model, but a direct comparison of the models is not provided and, as a result, the authors are not provided about a clear evidence of the effectiveness of one algorithm compared to the other. This point, if is believed to be addressed by authors, should be more significantly emphasized (and better described).

6) Discussion is too long and there is not a clear presentation and discussion about the limits. Please, revise this aspect.

6. PLOS authors have the option to publish the peer review history of their article (what does this mean?). If published, this will include your full peer review and any attached files.

Reviewer #1: **Yes: **GIOSIANA BOSCO

Reviewer #2: No

---

## [Author Response · Author response to Decision Letter 0]

17 Aug 2023

<Response to Reviewers >

Thank you very much for your careful review and advice on our research. All the authors highly respect the opinions of the reviewers, and all the authors have discussed deeply and tried to make modifications to reflect all the opinions of the reviewers. This is a study in the process of establishing a new model to confirm the risk of cardiovascular disease using big data of a Korean sample cohort. The revised manuscript was submitted as a track change version and a clean version. Responses are written in blue text color, and actual corrections are written in green text. Please check the revised contents and give a positive review result. Thank you again.

General Comment: 

In your Data Availability statement, you have not specified where the minimal data set underlying the results described in your manuscript can be found. PLOS defines a study's minimal data set as the underlying data used to reach the conclusions drawn in the manuscript and any additional data required to replicate the reported study findings in their entirety. All PLOS journals require that the minimal data set be made fully available. For more information about our data policy, please see http://journals.plos.org/plosone/s/data-availability.

- Answer: Thank you for your comment. We added the ‘Data availability’ section in the manuscript. Public sharing for our raw data is forbidden, and the data access is only allowed to registered researchers.

Therefore, the next contents have been added.

- Change in the manuscript:

Data availability

Access to the NHIS-NSC data is restricted to registered researchers, and public sharing of the raw data is not allowed. Detailed information about the data can be found at https://nhiss.nhis.or.kr/.

Comment (1-2)

1) Abstract. The introduction is poorly written and needs a very extensive rephrasing in order to provide to the reader a very straightforward message. Actually, there is not a clear rational about the gender-difference that is postulated in the heading and the design of the study and the analysis.

2) Abstract: conclusion section is even more elusive and very poorly clear. At the end the reviewer believes that the reader would not be provided of a clear message from this data. Please, provide a very important revision of the abstract and, consequently, of the rationale in the background/introduction section.

Answer: That's an important point. The abstract was completely revised. The results and conclusions are clearly written based on the research contents.

Comment (3)

3) Appendix 1. The descriptive table is not very helpful and actually the reviewer is not convinced about its added value. Vice versa, a table with descriptive statistics of the population strata according to CVD risk would be even more informative. In this way, this could be the first main table starting from the results section.

- Answer: Sorry for the uncertainty of the descriptive tables. We agreed that Appendix 2 (descriptive statistics of the population strata according to CVD risk) contains important information. In Appendix 1, We provided additional information by adding P-values generated by the t-test and the chi-square test, indicating the degree of sex difference for each variable. These tables (Appendix 1 and Appendix 2) were moved to the main part (Table 1-2) considering the importance of their contents. We also added the ‘Descript Statistics’ section in the ‘Result’, which provides brief descriptions of these tables.

Comment (4)

4) Table1 and Table 2. The way covariates are presented is not easy-to-get for the readers. Perhaps, some more explanation of how the models are conducted a which is the sense of each factor included in this table would be appreciated.

- Answer: Thank you for your comment. We’ve incorporated an explanation of the risk score calculations in the ‘Materials and Methods’ to aid reader comprehension. Table 1, which simply presents the list of parameters used in each model, has been relocated to the Supporting information section (S1 Table). 

- Furthermore, we’ve enriched the description of what was previously the Table 2, now Table 4. The model previously known as "KCDRS" has been renamed to "data-driven" to reflect its role more accurately. This ‘data-driven’ model was built to optimally fit our data, and is used as a gold standard to evaluate other models. Both FRS and the data-driven model identified Ln-treated SBP as the most influential variable. Given that the data-driven model highlighted blood pressure as the most critical risk factor in our study population, the model assigning the greatest weight to SBP (FRS) provided the most precise fit to the data.

Comment (5)

5) Figure 2, Figure 3, Table 3. Major point of criticism: actually, the authors are evaluating the areas under the curve of each model and how the performance is, for each single model, but a direct comparison of the models is not provided and, as a result, the authors are not provided about a clear evidence of the effectiveness of one algorithm compared to the other. This point, if is believed to be addressed by authors, should be more significantly emphasized (and better described).

- Answer: Thank you for your comment. To more directly illustrate comparisons between each model, we have restructured the ‘Result’ section with a more detailed explanation, and adjusted the tables and figures. In the revised version, both Table 3 and Figure 2 exclusively feature the results derived from the original FRS and ASCVD models. Table 3 displays the AUC values for the two models and the P-values obtained from AUC comparison. Additionally, chi-square values and P-values in Fig 2 come from the goodness of fit test, where a smaller chi-square value and a higher P-value suggest that the model provides a more accurate representation of the observed data. Fig 3 and 4 present consolidated results from both the Delong test and the chi-square test, comparing the original models with the new models. These results facilitate a comprehensive understanding and comparison of which model delivers superior prediction accuracy.

Comment (6)

6) Discussion is too long and there is not a clear presentation and discussion about the limits. Please, revise this aspect.

Answer: In the discussion, we identified and described the differences of our tool from existing tools such as FRS and the contents of other studies. In addition, redundant or repetitive background content was resolutely deleted. Some references have been removed. Please review the text. We were impressed by the reviewer's meticulous review. Thank you.

---

## [Decision Letter · Decision Letter 1]

12 Sep 2023

Sex Differences in the Applicability of Western Cardiovascular Disease Risk Prediction Equations in the Asian Population

PONE-D-23-01704R1

Dear Dr. Lee,

We’re pleased to inform you that your manuscript has been judged scientifically suitable for publication and will be formally accepted for publication once it meets all outstanding technical requirements.

Kind regards,

Chiara Pavanello

Academic Editor

PLOS ONE

Additional Editor Comments (optional):

Please address the additional minor comments of reviewer #2

Reviewers' comments:

Reviewer's Responses to Questions

**Comments to the Author**

1. If the authors have adequately addressed your comments raised in a previous round of review and you feel that this manuscript is now acceptable for publication, you may indicate that here to bypass the “Comments to the Author” section, enter your conflict of interest statement in the “Confidential to Editor” section, and submit your "Accept" recommendation.

Reviewer #1: All comments have been addressed

Reviewer #2: (No Response)

2. Is the manuscript technically sound, and do the data support the conclusions?

Reviewer #1: Yes

Reviewer #2: Yes

3. Has the statistical analysis been performed appropriately and rigorously? 

Reviewer #1: Yes

Reviewer #2: Yes

4. Have the authors made all data underlying the findings in their manuscript fully available?

Reviewer #1: Yes

Reviewer #2: Yes

5. Is the manuscript presented in an intelligible fashion and written in standard English?

Reviewer #1: Yes

Reviewer #2: Yes

6. Review Comments to the Author

Reviewer #1: The authors considered all the suggestions. The paper has been improved in each part. All the contents are now even more understandable. As I have already said, I found this study interesting as it provides a new cardiovascular risk score model which seems more suitable for Koreans, allowing sex differences in cardiovascular risk estimation and providing a practical solution.

Reviewer #2: 1) Abstract. The introduction is poorly written and needs a very extensive rephrasing in order to provide to the reader a very straightforward message. Actually, there is not a clear rational about the gender-difference that is postulated in the heading and the design of the study and the analysis.

2) Abstract: conclusion section is even more elusive and very poorly clear. At the end the reviewer believes that the reader would not be provided of a clear message from this data. Please, provide a very important revision of the abstract and, consequently, of the rationale in the background/introduction section.

R1: authors improved the sequence of concepts in the abstract, which now is much more informative.

3) Appendix 1. The descriptive table is not very helpful and actually the reviewer is not convinced about its added value. Vice versa, a table with descriptive statistics of the population strata according to CVD risk would be even more informative. In this way, this could be the first main table starting from the results section.

R1: OK. Actual Table 2 (=former Appendix 2) does not include p value comparing the incidence rates between men and women. Actually, it is not sufficient that the confidence intervals are all over 1 but it would be much more informative to had information regarding significant differences of the incidence rates between groups.

4) Table1 and Table 2. The way covariates are presented is not easy-to-get for the readers. Perhaps, some more explanation of how the models are conducted a which is the sense of each factor included in this table would be appreciated.

R1: the authors provided detailed replies. Please, turn “diabetes” in the table into “type 2 diabetes” (if authors do not have a separate information regarding type 1 diabetes).

5) Figure 2, Figure 3, Table 3. Major point of criticism: actually, the authors are evaluating the areas under the curve of each model and how the performance is, for each single model, but a direct comparison of the models is not provided and, as a result, the authors are not provided about a clear evidence of the effectiveness of one algorithm compared to the other. This point, if is believed to be addressed by authors, should be more significantly emphasized (and better described).

R1: The reviews thanks the authors for very detailed information and clarification.

6) Discussion is too long and there is not a clear presentation and discussion about the limits. Please, revise this aspect.

R1: discussion is not properly structured and balanced.

7. PLOS authors have the option to publish the peer review history of their article (what does this mean?). If published, this will include your full peer review and any attached files.

Reviewer #1: No

Reviewer #2: No

---

## [Editor Report · Acceptance letter]

20 Sep 2023

PONE-D-23-01704R1 

Sex Differences in the Applicability of Western Cardiovascular Disease Risk Prediction Equations in the Asian Population 

Dear Dr. Lee:

I'm pleased to inform you that your manuscript has been deemed suitable for publication in PLOS ONE. Congratulations! Your manuscript is now with our production department. 

Kind regards, 

on behalf of

Dr. Chiara Pavanello 

Academic Editor

PLOS ONE